# The Impact of the Loss of a Child in the Prenatal Period on Marital and Family Life and the Religiosity of Women after Miscarriage

**Aleksandra Kłos-Skrzypczak**

Faculty of Theology, University of Silesia, 40-007 Katowice, Poland; aleksandra.klos-skrzypczak@us.edu.pl

**Abstract:** Experiencing the death of a child is one of the most difficult things a person can go through. The situation of parents who have lost a child during the prenatal period is burdened with a social taboo. In the public sphere, it is often referred as the "secret problem of women". The aspect of religiosity is extremely important in experiencing mourning after a miscarriage. The study is of a theoretical–empirical nature. The purpose of this text is to emphasize the value of human life at every stage, including in the situation of miscarriage. Online questionnaire research was conducted on a sample of 77 women, supplemented by content analysis of three of the most popular virtual support groups for parents who have experienced child loss in Poland. The aim of the conducted research was to attempt to illustrate how women cope with the experience of miscarriage and how this experience affects marital and family relationships. The issue of faith and the depth of religious life were highlighted as elements that provide support to parents after miscarriage at various stages of mourning.

**Keywords:** miscarriage; family life; religiousness

## 1. Introduction

The conception of a child causes changes in marital and family life. A successful pregnancy and childbirth bring joy to parents and the entire family. However, not every pregnancy results in the birth of a healthy baby. There are often special situations, such as miscarriages or stillbirths. According to data published in the UNICEF report "A Neglected Tragedy: The Global Burden of Stillbirths", "A stillbirth occurs every 16 s somewhere in the world. This means that every year, about 2 million babies are stillborn (...) in 2019, an estimated 1.9 million babies were stillborn at 28 weeks of pregnancy or later, with a global stillbirth rate of 13.9 stillbirths per 1000 total births" (A Neglected Tragedy 2020). In Poland, approximately 1700 women give birth to a stillborn child each year, and over 40,000 pregnancies end in miscarriage. Spontaneous miscarriage occurs in about 10% of all pregnancies that have been diagnosed, but the actual number of miscarriages is much higher than estimated (Chazan 2003, p. 90).

Among types of miscarriages, the most common is spontaneous miscarriage, which is defined as "(...) the expulsion or extraction from the mother's body of a fetus that does not breathe or show any other signs of life, such as heart activity, umbilical cord pulsation, or clear voluntary muscle contractions, provided it occurs before the end of the 22nd week of pregnancy (21 weeks and 7 days)", whereas stillbirth is defined as "(...) the complete expulsion or extraction from the mother's body of a fetus, provided it occurs after the end of the 22nd week of pregnancy, and after such expulsion or extraction, the fetus does not breathe or show any other signs of life, such as heart activity, umbilical cord pulsation, or clear voluntary muscle contractions" (Journal of Laws 2022, item 1304). According to this provision, two conditions for miscarriage must be met: the appropriate gestational age and the expulsion or extraction of a dead child from the uterus (Guzdek 2023, p. 328).

Despite the passage of time and advancements in medicine, the death of a conceived child remains a socially indifferent fact: "the suffering of mothers who have lost their unborn baby is disregarded, the patients who lose a baby are treated like things, the experience of such a loss is totally dehumanized, and the loss and the associated problem are openly denied" (Guzewicz and Szymona-Pałkowska 2014, p. 79). According to Urszula Dudziak, the sadness caused by miscarriage generates a sense of homelessness for the child who has not become rooted in family structures yet, leaving no trace in their own home. The tragedy of this situation is intensified by the lack of mourning rituals and the avoidance of specialists from various fields in addressing this topic (Dudziak 2013, p. 184).

This text is intended to show how faith and religiosity affect marital and family life after the birth of a child in the prenatal period. The first part of the text presents the Christian dimension of the dignity of the human person from the moment of conception and what, in the understanding of the Catholic Church, the attitude of responsible parenthood is. Then, the issue of miscarriage is presented in medical and psychological aspects and, finally, the marital, family and religious aspects are emphasized. The third part of the text presents the methodology and results of the research. The final (fourth) part of the article presents the conclusions from the conducted research.

## 2. Life after Miscarriage

For parents, the loss of a child, regardless of the duration of pregnancy, is very painful. It becomes difficult to speak about the pain, sadness or grief following the loss of a no, that's the title of the blog posts. The bibliography includes the title along with a link to the page and the date of access.child during the prenatal period, as pro-choice communities regard pregnancy termination within the first 12 weeks as a "normal part of every woman's life" and assert that abortion should be approached "with care and respect". It is important to establish how to describe the void after a miscarriage, since, in the opinion of pro-choice supporters, we are dealing with a fetus or "very small cells connected to each other" (Rozmowa o aborcji z dzieckiem—jak to zrobić? 2023), which will only become a child in the perspective of time (though the timeframe remains uncertain) (Jak pisać o aborcji bez stygmy? 2023). This is why the Christian perspective on the value of human life in the prenatal period is extremely important and worth discussing.

### 2.1. Dignity of the Human Person from Conception

According to Christian ethics, human life is a fundamental value of a man as a person and occupies a central place in the entirety of goods, because "a person exists as a person, and if they were not a person, they would not be a human being. This means that being a person is not an additional quality to humanity, but an integral part of it" (Kieniewicz 2013, p. 26). The value of human life is not only an individual good but also the most valuable good of the entire human community. It should be remembered that a man is a social being and human social life is a corporate life. "Individuals are bound to families, social groups (...) leisure and school groups. Every person is born into a social group, acquires their first experiences there (...) and usually leaves this world in the context of collective experience" (Goodman 2009, p. 67). Human existence begins and develops in interpersonal relationships, within emotionally, socially and culturally close circles of individuals. From the moment of conception, a child should be identified as a member of a specific community and should be protected by that community, as "from conception, a child has an impact on the community to which it belongs" (Kornas-Biela 2002, p. 9).

The argument for the value and dignity of human life can be found in the Old Testament: in the Book of Genesis, in the commandment given by God to Noah's descendants after the flood (Genesis 9:6), or in the story of the Israelites' exodus from Egyptian captivity (Exodus 1:15–22), as "respect for life is, in a specific dimension, a sign of the absolute value of a man, who was created in the image of God" (Czym jest człowiek? 2019, p. 242). Concern for human life also stems from humanity's origin itself, as God, by giving life, guarantees its inviolability, even through the fifth commandment: "You shall not murder".

By endowing humanity with the dignity of God's Child and the prospect of eternal life, God reaches the pinnacle of the Christian truth about life (Ślipko 1995, p. 42).

In the encyclical Evangelium Vitae, Pope John Paul II recalls that "(...) Church has always taught and continues to teach that the result of human procreation, from the first moment of its existence, must be guaranteed that unconditional respect which is morally due to the human being in his or her totality and unity as body and spirit" (John Paul II 1995, p. 60). The Congregation for the Doctrine of the Faith states that man is entitled to a set of rights, including the first, the most important, constituting the basis and condition of the others, namely the right to life. "From the time that the ovum is fertilized, a life is begun which is neither that of the father nor of the mother, it is rather the life of a new human being with his own growth. It would never be made human if it were not human already" (Declaration on Procured Abortion 1974, p. 12). A human being, from the moment of conception, deserves respect for their dignity as an individual and for their sexuality (Donum Vitae 1987). While considering the dignity of the conceived child, two assumptions should be accepted: "firstly, that the child is a person, and secondly—the child is a person from the beginning of its existence, and therefore possesses human nature and dignity from the beginning, regardless of the phase of its development or health condition" (Imielska 2014, p. 53). From a Christian perspective, every human life is a gift and "it is perceived as a God's deposit placed in the hands of man, which should be given special care both in its beginnings and in its final moments of existence" (Stec 2016, p. 209).

### 2.2. Responsible Parenthood

In the Christian perspective, marriage is seen as a mutual self-giving of a man and a woman, and the act of marriage itself is treated as a way of mutual giving. The constitution "Gadium et spes" states emphatically that "children are really the supreme gift of marriage and contribute very substantially to the welfare of their parents" (Pastoral Constitution on the Church in the Modern World 1965, p. 50). The encyclical "Humanae Vitae" introduces the term "responsible parenthood", which is defined through the prism of biological, emotional, economic and religious aspects (Paul VI 1968, p. 10) and signifies the undertaking of the vocation to transmit life and education with an awareness of responsibility before the Creator (Bohdanowicz 2016, p. 182). The aspect of marital relationships is particularly significant during pregnancy. Pregnancy is a unique state, and waiting for the birth of a child causes reorganization of the marital system in every perspective of functioning, starting from a change in social status, through defining new social roles, to ending with a change in relations between partners. The area of interest of prenatal psychology is the importance of social relations, especially marital and family relations, in shaping the bond between parents and children. Strong marital relationships have a profound impact on the development of emotional attachment between parents and the child (Kucharska and Janicka 2018).

The progress of medical knowledge and technology, especially in the field of ultrasound scan, allows parents to participate more consciously and with greater engagement in all stages of a child's prenatal life. Celebrating pregnancy in public has become incredibly popular and widely accepted by society. In the era of social media, expectant parents eagerly share ultrasound images of their babies at specific stages of development, organize commemorative pregnancy photoshoots (known as "baby bump" photography) and hold baby showers—celebrations in honor of the mother, usually planned in the later stages of pregnancy. Society provides pregnant women with various privileges, including fully covered healthcare services, priority in public spaces and dedicated parking spots in urban areas. Local support groups for pregnant women are becoming more active, childbirth classes are thriving and improvements in healthcare services are becoming more common.

### 2.3. Interrupted Anticipation

A pregnant woman, regardless of the stage of her pregnancy, is treated by society as a "national treasure", the heart of her family and is legally, morally and socially protected.

So, who is the woman who has lost her child during the prenatal period? Who does she become when she is no longer carrying a child, but grief (Mirecka 2017, March 7)? What do you call parents who have lost their child as a result of miscarriage? How can you assist a woman and a man who have experienced a miscarriage? It is impossible to put into words the pain that parents feel at the moment of losing a child during the prenatal period because a miscarriage is like childbirth and experiencing the death of a loved one at the same time. The loss of a pregnancy is a painful and traumatic experience for parents, and the fact that the child's death occurs at an early stage of fetal life does not diminish the pain.

Importantly, "(...) parents who have lost their children due to miscarriage (...) experience not only pain and grief but also encounter many external difficulties, largely resulting from a lack of understanding of this particularly challenging situation" (Nowicka 2017, p. 147). In the case of a child's death during the prenatal period, especially before the 22nd week, medical staff often use medical terminology that suggests that the child never existed. In hospitals, terms like "dead pregnancy", "miscarried fetus" or "expelled mass of cells" are used, even though just a moment ago, parents were saying, "We heard the baby's heartbeat" and "We felt the baby's movements" (Kłos-Skrzypczak 2016, p. 81). The loss of a child during the prenatal period is a surprise to parents, and their reactions to this situation are incredibly strong and dynamic on a cognitive, emotional and behavioral level. Many parents discover the "value of the child, a real person who changed the perspective of their own future", at the moment of miscarriage (Łuczak-Wawrzyniak et al. 2010, p. 375).

## 3. Emptiness That Cannot Be Described with Words

Culturally, we have become accustomed to associating femininity with motherhood. Still, in the prevailing belief, the ultimate confirmation of femininity is giving birth to a child. A woman who cannot bring a child into the world breaks away from the family tradition, feeling isolated and unfulfilled. Miscarriage is the ultimate evidence of "procreative failure", and it is the most common reproductive problem encountered. As such, this issue needs to be considered from medical, social, familial and psychological perspectives (Napiórkowska-Orkisz and Olszewska 2017, p. 530). Miscarriage also has a physiological dimension, often uncomfortable, involving blood loss, lower abdominal pain and uterine contractions (Guzdek 2023, p. 335). All these processes also occur during childbirth but with a vastly different outcome. In the case of miscarriage, it is difficult to articulate the extent of suffering endured, only to ultimately lose one's child.

### 3.1. Miscarriage in a Medical Context

The experiences of women who have miscarried are incredibly difficult and painful. The intensity of women's psychological reactions at the time of miscarriage resonates in the long-term perspective, affecting relationships with their closest environment. The loss of a child in the prenatal period often triggers shock, anger, denial, helplessness and feelings of guilt. If we also consider the lack of empathy or the "inability to communicate medical information while respecting the dignity of the patient and her lost child" (Konarska 2023, March 31), the loss of a child in the prenatal period becomes a traumatic experience. According to Dorota Kornas-Biela, "experiences related to child loss have a processual nature, developing slowly, deepening, and fading away" (Kornas-Biela 2020, p. 336). During a woman's hospital stay after miscarriage, two contrasting attitudes often emerge towards the loss of the child: shock, disbelief, sorrow or sadness, which typically manifest through crying, and an attempt to take control over the situation and manage their own emotions (Kornas-Biela 1999, p. 181). The influence of inappropriate reactions from the surroundings, especially from doctors and midwives, is characterized by comforting statements like: "You are young to have another child" (Szymańska 2011, p. 103). Gynecologists might display a medicalizing attitude and assume the right to make decisions on behalf of the patient regarding the handing over of mementos related to the deceased child. Doctors, by not providing such mementos, argue that "the best method is forgetfulness" or "it only worsens

the problem" (Szymańska 2011, p. 104). For medical staff, miscarriage is often seen as an insignificant event that will not subject a woman to permanent psychological effects (Adolfsson 2006, p. 11). However, after experiencing a miscarriage, parents need healthcare workers to help them avoid further prolonged emotional implications. Clear and effective methods of communication by medical staff, as well as respecting the privacy of women after miscarriage, are intended to serve this purpose (Galeotti et al. 2022, p. 22).

### 3.2. Grief after the Loss of a Child

Reactions related to the loss of a child in the prenatal period are individual in nature. Many researchers argue that the intensity of grief, sadness or anxiety, which usually appear after a woman leaves the medical facility, is similar to what occurs after the loss of a loved one (Kornas-Biela 1999, p. 180). Miscarriage is a loss that deserves mourning, and its intensity varies depending on many factors, such as previous reproductive experiences, the desire to be a mother, having a child, marital and family support and personality traits. The source literature estimates that grieving should last from 12 to 18 months and involves five stages:

- Negation;
- Anger;
- Haggling;
- Depression;
- Approval.

The specified stages of mourning serve as a tool for identifying feelings that are difficult to precisely place on a timeline, as they are highly individual in nature (Kübler-Ross and Kessler 2005, p. 7). Research indicates that grief symptoms after a miscarriage occur in 90% of women (Białek and Malmur 2020, p. 135). Despite these facts, societal acceptance of grief after miscarriage remains low, as there is a tendency to label such grief as excessive. Consequently, there is a high barrier for grieving parents to seek help (Mergle et al. 2022, August 18). The period of mourning is a time when prospective parents face suffering, isolation, loneliness and seek answers to troubling questions. There is often no one to talk to about what they have experienced, and the closest environment, not knowing how to behave in such a situation, either remains silent or diminishes the experienced mourning. These situations are highly significant because unresolved or prolonged grief can lead to psychological disorders, such as depression, anxiety disorders or post-traumatic stress disorder (Bielecka 2012, p. 65).

### 3.3. Miscarriage and Mental Condition

Loss after a miscarriage can be observed in both women and men, but the intensity of emotions is often greater in women, and their experience tends to be more intense. According to Bożena Miernik, "the intensity of grief is related to the duration of pregnancy—the intensity of grief increases in women who have experienced a miscarriage later in pregnancy" (Miernik 2017, p. 258). Depressive symptoms are a common emotional reaction to the experience of losing a child in the prenatal period for women. Statistics show that the rate of depression among women who have experienced a miscarriage can reach up to 55%, and rates of anxiety range from 28% to 45%, both immediately after the event and even up to 6 months afterward (Białek and Malmur 2020, p. 135). The probability of depression increases proportionally with the number of miscarriages and the length of time waiting for offspring. About one-third of women diagnosed with depression after a miscarriage are at increased risk of suicidal thoughts. These findings are supported by research conducted in Finland from 1987 to 1994: the average annual suicide rate in the year following a miscarriage was significantly higher than that of women in the general population, at 18.1 per 100,000. The duration of pregnancy is of great importance in terms of the severity of symptoms, but it is not important at which week the miscarriage occurred; what matters is how the woman perceives the loss of the child. Interestingly, marriage has been identified as a protective factor against the aggravation of depressive symptoms after

the loss of a child in the prenatal period. Research indicates that married women report symptoms of depression less frequently than those in informal relationships (Białek and Malmur 2020, p. 136).

Acute stress syndrome and post-traumatic stress disorder (PTSD) can be observed in women who have experienced miscarriages. PTSD is a disorder that results from anxiety, which is a consequence and reaction to an extremely stressful event, including the loss of a child in the prenatal period. In December 2019, research conducted in three London hospitals was published. The aim was to assess the impact of early pregnancy loss on the emergence of anxiety, depression and PTSD. During the study, it was found that almost one-third of women (29%) experienced post-traumatic stress disorder compared to other healthy pregnant women. One in four women after a miscarriage (24%) felt anxious, and one in ten experienced depression. The study was repeated nine months after experiencing the loss of a child, and the conclusions were as follows: almost one in five women (18%) had post-traumatic stress disorder, one in six (17%) felt anxiety and one in twenty had depression (Pregnancy Loss Leads Post-Traumatic Stress in One in Three Women 2021, February 9). Conducted research confirms that as many as 77% of women after a miscarriage report experiencing intrusive memories, anxiety in specific situations, frequent nightmares and 68% of them describe a strong sense of helplessness (Białek and Malmur 2020, p. 137).

The loss of a child during the prenatal period affects a woman's somatic sphere, causing sleep disorders, loss of appetite, mood lowering and fatigue. Women after a miscarriage "show reduced interest in the sexual sphere, are irritable in social contacts and withdraw from them (...) they have a lower sense of self-worth and presence of suicidal thoughts" (Trębicka 2017, p. 247). The uncertainty experienced by women after a pregnancy loss contributes to a high level of anxiety, which can be a greater psychological burden than depression. This includes concerns about the return of the menstrual cycle, the desire to conceive another child and the risk of recurring miscarriages (Nynas et al. 2015, January 29). Depression and anxiety are common after a miscarriage, and the symptoms tend to persist for up to 30 months after experiencing the loss of a child.

### 3.4. Miscarriage and Marital Relations

Experiencing a miscarriage is incredibly difficult for both parents, as "not being able to know the face of their child means not being able to experience their identity and personal individuality. It is one of the most difficult sufferings (...) of parents who have lost their child before birth, especially in the early stage of their prenatal development" (Modlitewnik Rodzin Dzieci Utraconych 2018, p. 46). It is difficult to find information on the impact of miscarriage on the psychosocial condition of men in the literature. However, the father of the child, in the face of a miscarriage, also experiences sadness, grief and disappointment. He also goes through a process of mourning, even though these emotions are contrary to the accepted cultural norms. This is likely due to the fact that, for biological and psychosomatic reasons, the woman establishes the initial relationship with the child. Frustration, anger, guilt and reduced self-worth are common among men after a pregnancy loss. However, the grieving process is experienced differently by men, with less intense emotions. Perhaps that is why it is up to the man to support the woman, to provide her with a sense of closeness and to give her strength.

However, as research indicates, repeated miscarriages can lead to erectile dysfunction and anxiety about intimate relationships among men. Men who have seen the child during ultrasound examinations tend to have a harder time coping with the loss of a child in the prenatal period. Comparing the rate of depressive symptoms in 56 couples after a miscarriage, it could be concluded that 29% of women and 10% of men had an elevated level of depression within a week of the child's death (Białek and Malmur 2020, p. 136). It is evident that data concerning depression in men after miscarriage are underestimated. Often, the symptoms of depression in men include behaviors, such as anger, aggression, suppression of emotions, isolation, distraction, irritability, risky behavior or sleep disorders (Lewis and Azar 2015, p. 12).

It happens that in case of fathers, the consequences of losing a child in the prenatal period may appear with a delay and manifest in the form of risky behaviors, such as turning to intoxicants. Men can be overwhelmed by the way women experience grief, feeling unable to meet their expectations, leading to tension in their interactions that fosters conflicts and creates mutual distance. The disproportion in the intensity of grieving processes can make women feel isolated and misunderstood, contributing to the deepening of depression (Kornas-Biela 1999, p. 187). The conducted research shows that some men in the situation of miscarriage tried to "grasp the reality of the life that ended" (Williams et al. 2020, p. 138). They saw being a father as a possibility in an abstract future, not a certainty in the tangible present. They perceived the loss of the child as a loss of potential. Some research has shown that men consider miscarriage in biological terms, thus excluding any emotional involvement.

Marital relationships, in the face of miscarriage, are put to the test, difficulties arise and disruptions in communication occur. Mutual blaming, anger and lack of support from other family members can weaken the marital bond and often lead to the breakdown of the relationship. From conducted research, it is evident that couples who have experienced a miscarriage had a 22% higher likelihood of separation compared to couples who did not experience it. "Married couples who experience the mourning process in a mature way while simultaneously working on strengthening their marital bonds can actually become even stronger after such a trial" (Miernik 2017, p. 256).

Difficulties also arise in intimate relationships. Research conducted around 2020/2021 indicates that women after experiencing a miscarriage felt less valuable, and their sexual activity no longer brought them pleasure. The emotional state of women combined with physiological reactions to sexual stimuli has a significant impact on the quality of the relationship between partners (Białek et al. 2022, p. 813). The lack of formal identification of women in need of support in this area and, in appropriate cases, formal treatment can expose parents who have experienced a miscarriage to a significant worsening of symptoms, psychosocial impairment and the risk of miscarriage in the first trimester of any subsequent pregnancy.

According to Adriel Booker, although miscarriage is extremely difficult for marital relationships, they are not doomed to failure. Research indicates that if a woman feels that her partner is engaged in the process of grieving, sharing emotions and experiences, then the partnership becomes more enduring and stronger. The way in which spouses react to each other after a miscarriage has a direct impact on their relationships, including their intimate ones. As Booker points out, for many couples, a properly and well-lived mourning process after a miscarriage strengthens marital relationships and deepens their intimacy (Booker 2023). The sexual sphere is extremely important in partner relations. Miscarriage, as mentioned before, affects the sexual aspect for both partners. A common behavior among women is refraining from sexual intercourse due to a decrease in libido, issues with experiencing orgasm, pain resulting from medical procedures or difficulties in daily life interactions. The sexual dimension for women carries both physical and psychological significance. The decision to engage in sexual activity is driven by a desire for closeness, seeking security and acceptance. "Sexual life is a very important sphere, and the problems that arise in it can indicate the need for assistance for the partners on this level" (Trębicka 2017, p. 253).

*3.5. Miscarriage and Family Relationships*

The death of a child during the prenatal period has lasting psychological consequences, not only for the mother but also for her entire family. Not only is the father of the child a person on whom the mother strongly depends for emotional support but this issue also applies to the siblings of the child who died in the prenatal period. How children experience the loss of a sibling is influenced by their age, knowledge of prenatal development and the parenting style demonstrated by their parents. It is important at what stage of pregnancy children were informed about the conception of their sibling and whether they were

involved in establishing a connection with the unborn child through actions, like touching the mother's belly or talking to the baby (Miernik 2017, p. 263). If children were aware that their parents were expecting a new family member, the information about the death of their sibling should be presented in line with their cognitive and emotional capabilities. As Piotr Guzdek writes, "Hiding the fact of the death of an unborn brother or sister from the deceased's siblings carries negative consequences. Not knowing the actual reason for their parents' sadness, the living offspring may unjustifiably blame themselves for their parents' difficult psychophysical condition and potential conflicts" (Guzdek 2017, p. 390).

Children empathize with their parents' emotions, trying to find their place in a new and often challenging reality. It is not uncommon for siblings to feel anxiety, confusion or even a sense of being lost. They might experience disappointment due to the loss of their sibling, and if feelings of jealousy or fear of the new family member arise, they might attribute themselves as responsible for the situation. Older children might fear rejection or loss of love from their parents. Therefore, in the context of miscarriage, it is important to take the time to explain the reasons for the situation to the deceased child's siblings. Children should be relieved of the burden of loss, given sufficient attention and encouraged to talk about the emotions associated with the miscarriage (Guzewicz and Szymona-Pałkowska 2014, p. 93).

Indeed, this situation can be incredibly challenging for parents who have experienced a miscarriage, as they often find themselves unsure of how to help and support those in mourning, including family members. Therefore, after a miscarriage, parents must face not only their own pain at various stages of mourning but also focus on the experiences of the deceased child's siblings. This requires a high level of emotional maturity, self-awareness of their emotions and effective communication between parents. Open and honest conversations, as well as dedicating time to answer questions or doubts from the siblings, provide an opportunity for them to navigate the crisis and emerge stronger. Research indicates that parents offer support to their deceased child's siblings in three areas: "acknowledging and recognizing the child's grief, involving them in family rituals related to saying goodbye to the deceased child, and maintaining the family's memory of the child" (Miernik 2017, p. 263).

It is important to emphasize that comprehensive support provided to parents who have experienced the loss of a child in the prenatal period contributes to a proper process of grief among all family members. Therapeutic assistance should be extended to all family members, as it not only helps them better understand the dynamics between them but also enables each member to develop coping mechanisms for the challenging experience of miscarriage (Dziedzic 2022, p. 208).

### 3.6. The Experience of Losing a Child and Devotion

Culture, language and religion have a significant influence on the way grief is experienced after the death of a child in the prenatal period. It is through religion, among other factors, that the tradition of a proper approach to the issues of death or religious ceremonies is culturally rooted. In our culture, a funeral is a natural consequence of a person's death, an expression of respect that should be paid to the deceased individual. Similarly, this applies to a child who has passed away before birth: the decision about burial lies within the purview of the parents, who, in accordance with their worldview, may choose to bury their child but are not obliged to. The funeral of a miscarried child, which is granted "due to the inherent ontological identity possessed by the child during its life and the reverence owed to human remains" (Guzdek 2017, p. 390), serves as an excellent educational tool for close or distant bereaved family members. The burial of a child who died in the prenatal period has both micro and macro social dimensions: it shapes the attitude of responsibility towards conceived life, verbalizes the dignity of the human person and the burial site (grave) becomes a place of reflection and remembrance, a visible symbol of parents' yearning for the departed, unknown and unsoothed child.

The religious aspect can have a significant impact on the experience of grieving and can serve as a buffer that makes readjustment after a miscarriage possible. Faith and strong religious convictions, along with social support, contribute to a lower level of grief stemming from the death of a child in the prenatal period. Moreover, religious institutions are centers with rich resources of social support, and the participation of parents in religious practices helps in accepting this support at various stages of mourning (Allahdadian and Irajpour 2015, December 30).

Religious matters also have an impact on coping with psychological issues after a miscarriage. Specifically, the stronger an individual's religious beliefs are, the lower the level of hopelessness and grief experienced after the loss of a child in the prenatal period. Moreover, in highly developed countries where women are often identified by their ability to bear children, the loss of a child in the prenatal period can lead to social exclusion. Therefore, the support and openness of religious institutions at this time become invaluable. There is also a connection between the mental well-being of women after a miscarriage and their participation in religious services. Mothers who regularly participate in religious practices show lower levels of post-miscarriage depression compared to non-practicing women. Taking into account the positive influence of religion on particular stages of mourning after a miscarriage, it should be made easier for orphaned parents to participate in religious ceremonies (such as funerals, worship services and regular prayers). Religious beliefs and a rich spiritual life significantly influence how women experience the loss of a child in the prenatal period. Women's spiritual lives, according to Felicity Agwa Kalu, have a significant impact on finding meaning in miscarriage and the purpose of life. Some families grieving the loss of a prenatal child have found comfort by defining the meaning of their miscarriage within their religious and spiritual beliefs (Kalu 2019).

The role of a priest seems extremely important in this situation, whose words, spiritual consolation or confession can give parents comfort in mourning, filling them with faith in eternal life and meeting their unborn children in Heaven. As we read in the Letter to the Romans: "Hope does not disappoint, because the love of God has been poured out into our hearts through the holy Spirit that has been given to us" (Letter to Romans 5:5 1966).

## 4. The Methodology of the Conducted Research

To cope with the difficult experience of losing a child in the prenatal period, many parents turn to online support groups. Social media platforms provide a variety of groups where parents can easily and quickly register, share their experiences, receive support and find understanding.

*Data Collection*

For the purpose of this study, content from posts published between 1 April 2023 and 31 July 2023 was analyzed on three of the most popular online support groups for parents who have experienced loss, accessible through the social media platform Facebook.

1.　Child Loss (from Polish: Strata dziecka[1]): 1790 members, a group established 9 years ago (referred to as Group A);
2.　Heavenly Children's Parents (from Polish: Rodzice niebiańskich dzieci[2]): 5805 members, a group established 8 years ago (referred to as Group B);
3.　Child Loss—Orphaned Parents—Help and Support (from Polish: Strata dziecka—Osieroceni rodzice—Pomoc i wsparcie[3]) 5337 members, group established 5 years ago (referred to as Group C).

The posts considered were those posted within a specific time frame, and their content was related to the death of a child during the prenatal period (on average 70% of the content). To become a member of these groups, individuals need to send a request to join to the group administrator. Each of the mentioned groups has clearly defined rules of operation in their "Information" section: "We are here to support each other, give strength, and share experiences. In the group, we do not judge (...) we trust each other (...) we all have

a very positive attitude towards each other because we have an abundance of suffering" (Group B information). The administrator of another group mentions that "membership in this group requires mutual trust. Authentic, expressive discussions enrich the groups, but sensitive and confidential topics can also be addressed. Information shared in this group should remain within the group" (Group C information).

In the mentioned groups, a proprietary questionnaire on the topic of "The Impact of Child Loss on Marital and Family Relationships" was also shared. The questionnaire was made available through Google Forms survey software from 19 June 2023 to 14 September 2023. The research had a quantitative nature and was fully anonymous. The questionnaire only included questions regarding the respondent's age and gender. It did not require logging in from respondents. The questionnaire consisted of 13 questions: 12 closed questions with disjunctive and conjunctive cafeteria and 1 open question. Additionally, filtering questions (conditional questions) were applied, allowing the questionnaire to be structured in a way that respondents could answer only questions relevant to their situation. The estimated time for completing the survey was a maximum of 5 min.

In summary, responses were received from 77 respondents, all of whom were women. The age of the respondents ranged from 23 to 49 years. The largest group of respondents fell between the ages of 27 and 30, as well as between the ages of 32 and 35. These conclusions were based on the questions included in the survey questionnaire, which asked directly for gender and age.

## 5. Materials and Results

The conducted questionnaire study revealed that almost 43% of the miscarriages occurred after the 20th week of pregnancy, more than 36% of the losses took place between the 12th and 19th week of pregnancy, while the remaining almost 21% were within the first 12 weeks of pregnancy. These conclusions were based on the first question in the research questionnaire: "In what week did the miscarriage occur?" It is worth noting that each respondent provided a specific date (day, month, year). The time range covers almost 11 years: from 4 October 2012 to 31 August 2023. Next, respondents were asked a question about "How many miscarriages have you had?" More than three-quarters of the participating respondents experienced one miscarriage (73% of responses), 18% of the participants mentioned having two miscarriages, 5% of the respondents have experienced three miscarriages, while almost 3% of women reported having more than three miscarriages.

The above statistical data are supplemented by an analysis of the content of posts published on support groups (Groups A, B and C). Parents commonly use various terms to refer to their unborn children who passed away, such as "my son", "my daughter", "our Angel", "our Sunshine", "beloved Little Angel", "unborn son" and "our greatest treasure". It happens that children who died in the prenatal period had the name that their parents use in their posts ("son Maksiu", "darling Mati", "daughter Lenka").

The questionnaire-based research indicates that the questions most frequently asked by women after experiencing a miscarriage were as follows: "Why me?", "Why my child?" and "Why did this happen to me/us?". It is worth noting that many women (almost 30% of responses) blamed themselves for their child's death, posing questions, such as "What did I do wrong?", "How could I have let this happen?", "Why didn't I fight more with the doctor?" and "Why didn't I trust myself more than the doctors?".

Analyzing the results of the conducted research, it can be concluded that 87% of respondents declared themselves as religious individuals. Among those who answered positively to the question "Are you a religious person?", a follow-up question about the method of burying the child was posed. From the acquired information, it can be inferred that 61% of women opted for the burial of the child in an individual or family grave, while more than 13% chose burial in a collective grave for lost children. Almost 26% of respondents indicated that the hospital organized the burial.

It should be emphasized that the analysis of the content from support groups (mainly Groups A and B) indicates that parents regret the lack of opportunity to say goodbye to

their lost child and the fact that they did not decide on its burial. There are entries full of doubts of parents whose children were buried by the hospital with the following content:

> "Recently, I've been having thoughts whether my little one (...) was actually buried? I never saw my child. I asked at the hospital if I could see them, but they refused. I asked at the funeral home, I even cried and begged to see them. Unfortunately, they kept refusing. They said that they just take them from the hospital and put them in a coffin. I didn't even get to choose the coffin" (Group B, posts from 17 June 2023)

Believers who decided to bury their child were asked the following question: "Who participated in the funeral?" From the conducted research, it is evident that 74% of individuals who chose to bury their child (whether in an individual/family grave or in a collective grave for lost children) opted for a prayer at the gravesite or in the cemetery chapel. Just 26% of women affected by the loss of a child in the prenatal period chose a funeral with a funeral mass. Almost half of the funerals for the lost children (46% of responses) were of a familial nature—parents, grandparents and siblings attended the funeral. Over 22% of respondents indicated that distant relatives also participated in the funeral, and 32% of women indicated that the funeral ceremony was personal in nature: only the husband/partner and herself participated.

The content analysis of the posts indicates that parents of children who passed away in the prenatal period attach great importance to matters related to the burial of their child. Mainly users of Group A share designs of gravestones, photos of adorned graves of their unborn children and exchange information on where to buy personalized, named decorations for the grave (wreaths, vases, candles, toys). A lot of content is about which type of flowers is the most durable, how to decorate the grave for specific occasions and holidays, etc. Parents also inquire about epitaphs found on the graves of children who passed away before birth, especially users from Group B. It is worth mentioning the content of a few of them:

- "I'm not asking you God why you took him from us, but thank you for giving him to us";
- "You are nowhere, because you are already everywhere";
- "What a great treasure this grave hides only mom, dad and God know";
- "Rest in God dear Eryk, you will never be forgotten by us";
- "You are our part of Heaven...only time separates us";
- "You left a lifetime too soon";
- "Jesus, I place my treasure under Your cross";
- "Some babies learn to walk, others get wings right away".

In the questionnaire study, over half of the women, when asked the question "How did the loss of your child affect your relationship with your husband/partner", stated that their relationship had strengthened (over 44% of responses). Almost 12% of women admitted that experiencing the loss of a child in the prenatal period had a negative impact on their marital/partner relationships, leading to the dissolution of the relationship. Slightly over 44% of respondents stated that miscarriage did not affect their marital/partner relationships in any way.

In support groups for parents who have experienced loss, much attention is given to mental well-being. Women share songs that lift their spirits, poems dedicated to their unborn children and even therapeutic stories written after the loss (Group B, post from 22 June 2023). On each group, information about local support groups is shared, both those that meet in person and those held online. Many women also offer to engage in conversations with other women outside the group, either through social media or over the phone.

Respondents were also asked the following question: "How did your husbands/partners react to the miscarriage?" The women participating in the study stated that they received support from their husbands/partners at every stage of their grieving process (62% of

responses). They mentioned that their husbands/partners also experienced the loss deeply (over 4% of responses). However, 26% of women declared that after miscarriage, husbands/partners were restrained in expressing their emotions; they were looking for substitute forms of activity (almost 12% of responses) and indulging in addictions (almost 9% of respondents).

Then, the people participating in the study were asked the following question: "What forms of support did you use after the miscarriage?" From the questionnaire study, it is evident that more than 70% of respondents sought various forms of support after experiencing a miscarriage. This study also indicates that almost 70% of women who reported an improvement in their marital/partner relationships after a miscarriage broadly understood support: almost 29% of respondents went for individual therapy or couples therapy (more than 9% of responses). The influence on marital/partner relationships after a miscarriage was also attributed to support from a religious figure (discussions, confession)—this declaration was made by over 19% of women. A significant number of women admitted to seeking support from other women who had experienced miscarriages (39% of responses) or from support groups for women after child loss (20% of respondents). The support of parents/closest family members seems to be extremely important in the experience of losing a child in the prenatal period, with over 45% of women making this declaration. It is worth emphasizing that women usually use several forms of support at the same time.

## 6. Conclusions

The conducted comprehensive analysis of content related to prenatal child loss, supported by questionnaire studies and content analysis of posts in various support groups, enables us to formulate several key conclusions:

- Parents who have experienced loss acknowledge the full dignity and inherent value of their unborn children. This understanding of dignity is expressed through the use of personal pronouns, gender-specific terms or specific names or nicknames when referring to the children who have passed away before birth.
- Women who have experienced a pregnancy loss often struggle with intrusive thoughts and feelings of guilt, which constitute significant stages of the grieving process.
- The behavior and attitude of a doctor in the face of pregnancy loss have a significant impact on the mental well-being of parents.
- Parents go through the difficult period of mourning after the loss of a child in the prenatal period much more easily if they can organize a funeral for them. Typically, this is a small ceremony, a symbolic farewell, which, nonetheless, holds significant meaning for the parents' emotional state.
- Religious and spiritual beliefs enable individuals to find meaning in the loss of a child during the prenatal period and aid in coming to terms with miscarriage. Therefore, it is important for parents who have experienced miscarriage to utilize the full potential of their religious and spiritual convictions as resources for coping with the loss from a psychosocial perspective.
- In the face of child loss, women eagerly seek support from close family members, friends, as well as other women who have experienced miscarriage. Virtual support groups, often found on social media platforms or online forums, allow for connections to be formed with individuals who have undergone similar tragedies.
- There is still limited knowledge about how men experience grief after the loss of a child. A man who has experienced the death of a child in the prenatal period must confront not only his own emotions but also the suffering of his wife/partner.
- The aspect of faith influences marital and family relationships after miscarriage. Believing women declare an improvement in their marital relationships after experiencing the loss of a child in the prenatal period.

It is worth constantly emphasizing the great value of every human life, from the moment of conception. Pope John Paul II proclaimed that the dignity of a human person can be defended when it is considered inviolable from conception to natural death, and

"care for a child (...) is the first and fundamental test of the relationship between man and man" (Św. Jan Paweł II obrońcą życia 2023, March 10).

**Funding:** This research received no external funding.

**Institutional Review Board Statement:** Ethical review and approval were waived for this study, due to the fact that the research was fully anonymous and no sensitive date were collected. Consent to conduct the research has been obtained from the Institute of Theological Sciences of the University of Silesia in Katowice (Poland). Issue number: WTL/INT.476.3.2022. Date of consent: 7 December 2022.

**Informed Consent Statement:** Informed consent was obtained from all subjects involved in the study.

**Data Availability Statement:** The data presented in this study are available on request from the corresponding author.

**Conflicts of Interest:** The author declares no conflict of interest.

## Notes

[1]   Available online: https://www.facebook.com/groups/702835643062565/about (accessed on 28 July 2023).

[2]   Available online: https://www.facebook.com/groups/1553595081518532/about (accessed on 28 July 2023).

[3]   Available online: https://www.facebook.com/groups/1851271521785003/about (accessed on 28 July 2023).

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
