# Peer review of "The Impact of the Loss of a Child in the Prenatal Period on Marital and Family Life and the Religiosity of Women after Miscarriage"

_religions, doi:10.3390/rel14111434_

Round 1

Reviewer 1 Report

Comments and Suggestions for Authors

The Author very well justifies the first part of the article's topic, i.e. ‘The impact of the loss of a child in the prenatal period on marital and family life’. He presents original research and formulates logical and useful conclusions. The second part of the topic: ‘the religiosity of women after miscarriage’ is good and enough but less detailed. It is worth examining not only the sociological but also the theological dimension of women's religiosity in subsequent research. For example, examine how the image of God changes during the entire mourning process and whether it is consistent with the image of God contained in the Bible. A valuable element of the article is the description of the man's situation after a miscarriage. It is worth the author's next article to dedicate only to this issue, because, as He writes: „It is difficult to find information on the  impact of miscarriage on the psychosocial condition of men in the literature.” (cf. verse 291-292). Therefore, the author's research on men's mourning after a miscarriage will help in providing men with appropriate support.

Author Response

Thank you very much for your provided positive review. I am grateful for your interest in my article and for reading it carefully, as evidenced by the review sent. Thank you for your tips on conducting further, in-depth research on the topic of prenatal child loss. 

Reviewer 2 Report

Comments and Suggestions for Authors

A very high-quality text. A much-needed topic, although rarely discussed. The bibliography is correctly selected, and articles in English are valuable. The research conducted is extremely valuable, especially the analysis of support groups available online.The author moves very well in the issue discussed, and shows her erudition in the topic discussed.

One critical remark. The literature could be expanded to include texts in other foreign languages to make the article verifiable by more readers.

Author Response

Thank you very much for the positive review you provided. I am grateful for your interest in my article and careful reading, as evidenced by the review sent. Thank you also for all your comments and suggestions.

Reviewer 3 Report

Comments and Suggestions for Authors

POSITIVE ASPECTS OF THE TEXT:

1. It is appropriate that the article is set in a theological context, which the author emphasises, the numerous references to the teaching of the Catholic Church confirm this.

2. The sociological analysis carried out is extremely valuable, making the embedding of the topic in the present day evident. The author emphasises the topicality of  own research, this adds to the timeliness of the text.

3. It is excellent that the author points out the problem of fertility in modern times.

4. The author's presentation of the impact of miscarriage on marital and family relations enriches the article and makes it a part of the contemporary current of family studies, which is just being formed in Poland.

5 The methodology of own research was presented in an exemplary way. The research group was clearly defined, the methods used and the anonymity of the research. Filtering questions were used to control the research conducted. The research is fully professional.

6 The conclusions drawn are pertinent and presented in a synthetic manner.

COMMENTS

1) It is worth noting in the introduction that this is a classification of miscarriages included in Polish legislation.

2) It is worth enriching the article with a bibliography beyond English and Polish.

3) It is possible to expand the research group to include other groups; this is a suggestion for future publications by the Author.

4. From the sociological point of view, it is worthwhile to use in-depth research, which would allow to show other problems not only on the religious but also on the spiritual level.

Author Response

Thank you very much for the positive review you provided. I am grateful for your interest in my article and careful reading, as evidenced by the review sent. Thank you also for all your comments and suggestions. Any comments will also be introduced at the stage of further, planned research on the subject of prenatal child loss.

Reviewer 4 Report

Comments and Suggestions for Authors

The author's article addresses an important and timely topic. The text is divided into four main chapters and final conclusions. In part one, the author discusses the Christian dimension of the dignity of the human person from the moment of conception and the attitudes of responsible parenthood as seen by the Catholic Church. In the second part, the author describes the problem of miscarriage in medical and psychological aspects, then emphasized marital, family and religious aspects. The author also points out the importance of bereavement. Then he characterizes the methodology of the results of the research undertaken, and in the last part he presents the results obtained from the research conducted.  The topic of the article is extremely timely and important. The problems presented in it concern an important group in society. Issues related to the loss of a child concern a difficult, but constantly relevant topic. Therefore, it is necessary to undertake reflections on this issue. The methodology used does not raise any objections. The conclusions drawn from the conducted research are clearly presented by the author.
  The structure of the reviewed article is thoughtful and clear to the reader. The article meets the requirements of a scientific text. The work is original and worth undertaking research on a wider scale.

Author Response

Thank you very much for the positive review you provided. Thank you also for all your comments and suggestions. Any comments will also be introduced at the stage of further, planned research on the subject of prenatal child loss.

Reviewer 5 Report

Comments and Suggestions for Authors

I really appreciated the focus on the sanctity of life, especially from a Catholic perspective. This provides an important focus on understanding the impact of the loss on Christians. I recommend the authors revise the methods and results sections. I appreciate the "mixed-methods" approach taken in using a survey and completing a content analysis. In the methods section, I would like to see the actual survey. What questions were asked? I know this is a social survey and not a questionnaire with psychometric properties, but the information would aid the reader in understanding the information received. Second, the methods should describe the content analysis frame used. Margrit Schreier has an excellent text on conducting a qualitative content analysis. Adopting a formal method for analyzing the content would add to the validity of the results. In the results section, descriptions and themes should be more clearly developed. More direct connections are needed with the Christian spiritual aspects of coping with the loss of a baby prior to birth. 

Author Response

Thank you very much for your review and comments. I appreciate the substantive analysis of methodological issues included in the work. 

The comments included in the review have been implemented, as shown by the revised version of the text that I have attached to the Academic Editor.

In the course of further, extended research on the loss of a child in the prenatal period, the reviewer's comments will also be applied.

Round 2

Reviewer 5 Report

Comments and Suggestions for Authors

This is definitely an improved manuscript, however, it is not yet ready for publication. First, I would ask the authors to include the survey as an appendix for the readers to have access to. This would provide greater context for the types of questions posed to the groups. Second, more attention is needed on integrating qualitative analysis of the support groups' posts with the survey. One way to do this, which was done to a minimal extent already, is to combine specific questionnaire items with specific content from the groups' posts. For example, more information is needed discussing the themes that emerged for the concerns regarding the impact the miscarriage had on the spouse (husband) with relevant items from the survey. This provides context for the content analysis and the survey. Currently, the results/discuss read as a list of either content analysis or responses to the survey. Integrating these two separate streams of data demonstrate a more comprehensive mixed-methods approach. 

Author Response

Thank you very much again for your reply. In accordance with the instructions, I have prepared a survey questionnaire attached. The question dump is not as transparent as in the system. However, for those interested, I can provide a link to the research and, of course, access to the full results.
The reviewer's comments about the husband are very important to me. However, it is difficult to make this analysis based on the collected materials. The analyzed groups are dominated mainly by women (which was also confirmed by research). However, the reviewer's tips are extremely valuable to me, they constitute an inspiration for further research, which will be dedicated to men (husbands/partners) with the experience of child loss, using a mixed method.
